# ATTENTION ON ABSTRACT VISUAL REASONING

## ABSTRACT

Attention mechanisms have been boosting the performance of deep learning models on a wide range of applications, ranging from speech understanding to program induction. However, despite experiments from psychology which suggest that attention plays an essential role in visual reasoning, the full potential of attention mechanisms has so far not been explored to solve abstract cognitive tasks on image data. In this work, we propose a hybrid network architecture, grounded on self-attention and relational reasoning. We call this new model *Attention Relation Network* (ARNe). ARNe combines features from the recently introduced Transformer and the Wild Relation Network (WReN). We test ARNe on the Procedurally Generated Matrices (PGMs) datasets for abstract visual reasoning. ARNe excels the WReN model on this task by 11.28 ppt. Relational concepts between objects are efficiently learned demanding only $35\%$ of the training samples to surpass reported accuracy of the base line model. Our proposed hybrid model, represents an alternative on learning abstract relations using self-attention and demonstrates that the Transformer network is also well suited for abstract visual reasoning.

## 1  INTRODUCTION

Psychological models of human intelligence identify different manifestations of intellect. FLUID INTELLIGENCE describes the ability to adapt to new problems and situations without relating to previous learning outcomes (Hagemann et al., 2016). This ability is considered as one of the most important aspects for learning and is essential for solving higher cognitive tasks (Jaeggi et al., 2008). In order to excel in this type of intelligence, cognitive capabilities such as figural relations, memory span and inductive thinking are decisive. Fluid intelligence also paves the ground for ABSTRACT REASONING, the ability to use symbols instead of concrete objects. Furthermore, empirical data show that attention and fluid intelligence are strongly interlinked (Stankov, 1983; Schweizer, 2010; Ren et al., 2013). Subjects that perform bad on attention tasks are also more likely to show deficits in abstract reasoning and fluid intelligence (Ren et al., 2012; 2013).

RAVEN'S PROGRESSIVE MATRICES (RPM) is an established test method for intelligence, especially fluid intelligence and abstract reasoning (Bilker et al., 2012). It is a set of non-verbal tests showing several geometric objects arranged to a certain implicit rule (Figure 1a shows an example). The test subject has to complete the $3 \times 3$ matrix by picking an object that matches the implicit rule. The PROCEDURALLY GENERATED MATRICES (PGMs) (Santoro et al., 2018) dataset is motivated by RPMs and synthetically generated for training neural networks. Its core feature comprises several relations between objects and one of their attributes, e.g. the number of an object of a certain type in each matrix panel. A single matrix can contain up to four such relations simultaneously. A PGM's context and possible answers are shown in Figures 1a and 1b, respectively. Importantly, the PGM matrices are presented to a neural network as images such that the task requires a certain level of understanding of geometrical relations. The Wild Relation Network (WReN) uses different combinations of matrix elements as a mechanism for relational reasoning and is currently among the best performing solutions for the PGM dataset. However, despite the close relationship between attention and reasoning, the WReN is lacking a mechanism for attention.

Apart from the visual domain, it was recently shown that attention is also essential for another cognitive task that fundamentally relies on abstract reasoning: *language understanding*. The Transformer, an attention-based neural network, was introduced to improve machine translation and transduction (Vaswani et al., 2017). The Transformer embodies a self-attention mechanism to relate parts of its input with each other for writing output. Since this form of attention and abstract reasoning is

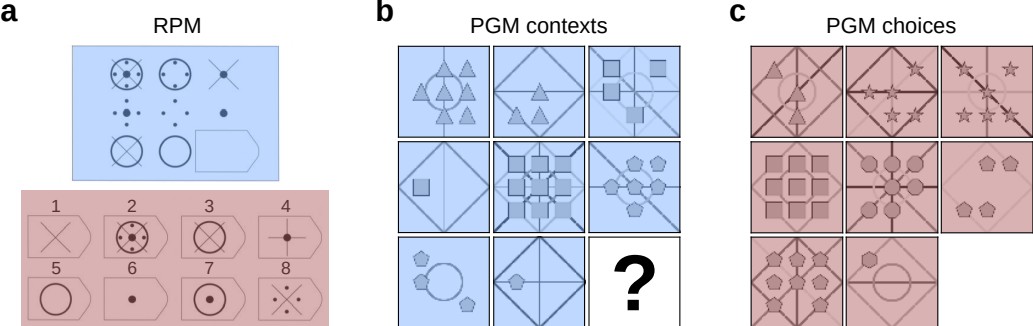

Figure 1: **Datasets for visual reasoning. a:** Sample from the RPM dataset (Bilker et al., 2012). Context: blue, choices: red. Correct choice: 1. Implicit rule: Subtraction along rows and columns. **b, c:** Sample from the PGM dataset (Santoro et al., 2018), contexts (**b**) and corresponding choices (**c**). The **?** in (**b**) should be replaced by a correct choice from (**c**). Correct choice: The hexagon in the last tile. Implicit rule: progression of shape types (number of edges) in each column. Colors added to enhance visualization.

vital for fluid intelligence we hypothesize that a Transformer augmented with relational reasoning capabilities can perform well on the PGM task.

Another special trait of human intelligence is the ability to infer rules with little supervision. Often, only few samples suffice for people to grasp the idea of tasks such as visual reasoning. In contrast to that, current deep network architectures are trained on huge quantities of data, e.g. the PGM dataset contains more than 1.2 million labelled samples. To see if self-attention also enhances this capability of deep networks we investigate the sample efficacy by training our proposed model only on a fraction of the total available data samples.

Our contribution to the field of machine learning is three-fold:

- We introduce the Attention Relation Network (ARNe) that combines features from the WReN and the Transformer network and can be directly trained on visual reasoning tasks.

- We evaluate ARNe on the PGM task and show that it significantly outperforms the current state-of-the art (Steenbrugge et al., 2018) by 11.28 ppt.

- We demonstrate that ARNe is very sample-efficient and achieves its peak performance with only 35% of the full PGM dataset.

This paper is organized as follows. In Section 2 we discuss related approaches and training datasets. In Section 3 we introduce the ARNe architecture and in Section 4 we show our simulation results. We conclude in Section 5.

## 2 RELATED WORK

**Language Modelling**  Language modelling, i.e. predicting the likelihood of future words given a set of contextual words, is a core task in natural language processing (NLP). While neural approaches to language modelling have some tradition (Bengio et al., 2003), the success of the word2vec (Mikolov et al., 2013) has revived and boosted the interest in neural networks for this task. While word2vec defines fixed vectors for words, recently, several approaches were proposed that condition word representations on their context (Howard & Ruder, 2018; Devlin et al., 2018; Peters et al., 2018). In particular attention-based models have emerged as the standard tool for language modelling as well as several downstream tasks, such as question answering (Rajpurkar et al., 2016) and translation (Vaswani et al., 2017). Starting with the seminal Transformer (Vaswani et al., 2017), follow-up models were successively improved (Devlin et al., 2018) and trained on larger text datasets achieving remarkable success across various NLP tasks (Radford et al., 2019). Besides the work in language modelling, in concurrent work, Transformer-based models have been employed for visual question answering (Li et al., 2019). In this work we take inspiration from the model

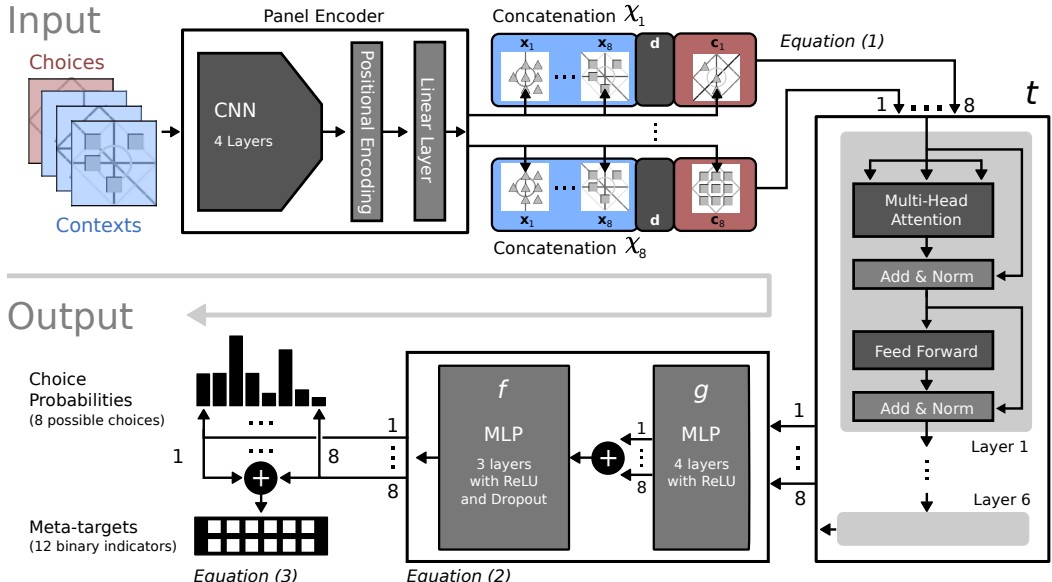

Figure 2: Overview of the ARNe network for abstract visual reasoning.

design of the attention Transformer. However, instead of applying it in a language modelling task, we use it for abstract reasoning.

**Abstract Reasoning**    Recently, there has been growing interest in abstract reasoning indicated by the introduction of datasets operating on textual data, e.g. babI (Weston et al., 2015), as well as on visual data, involving CLEVR (Johnson et al., 2017), FigureQA (Kahou et al., 2017) and NLVR (Suhr et al., 2017).

While classic recurrent neural networks have often been used for these tasks, it has been found that novel architectures such as attention and memory are better suited to address abstract reasoning. Attention mechanisms are a method to simplify complex information by attending to different locations of the given data. This is an advantageous procedure to let a neural network determine a specific reasoning operation on these selected locations. Memory is needed to preserve information in tasks where multiple consecutive reasoning operations are required. In LSTM (Hochreiter & Schmidhuber, 1997) for example, the hidden state is protected by gates from being overwritten by new input. Models incorporating attention and memory involve the Neural Turing Machine (Graves et al., 2014) which evolved into the Differentiable Neural Computer (Graves et al., 2016). Dynamic Memory Networks (Xiong et al., 2016) address both, visual and textual question answering. The recently introduced Memory, Attention and Composition (MAC) Network Hudson & Manning (2018) achieved excellent scores in the CLEVR benchmark by using multiple explicit reasoning steps. Recently, a reasoning module relying on attention Zambaldi et al. was designed to be employed in a reinforcement learning framework.

Our work differs by measuring fluid intelligence by understanding visual patterns rather than solving referential expressions (to which part of an image belongs a certain word) which is required in CLEVR, FigureQA, NLVR and many visual question answering tasks. The former task is considered in (Santoro et al., 2018) and (Steenbrugge et al., 2018) with which we compare our proposed method.

## 3    ATTENTION RELATION NETWORK (ARNE)

In this section we describe the Attention Relation Network (ARNe), which combines techniques from language modelling to abstract reasoning. It takes eight contextual panels and eight choice panels from the PGM, of which one fits to abstract relational rule implicitly determined by the context panels. The model output are logits of a corresponding fitness probability for each choice

Table 1: Modules and their corresponding parameters of ARNe.

| Module | | | Parameters |
|---|---|---|---|
| **Panel Encoder** | | | |
| CNN | $\left\{\begin{array}{l}\text{2D-Convolution}\\ \text{Batchnorm}\\ \text{ReLU}\end{array}\right.$ | $\Big\}$x4 | $3 \times 3 \times 32, \text{padding} = 1, \text{stride} = 2$ |
| Linear layer | Embedding | | $\frac{\text{H·W}}{256} \cdot 32 + 9 \times 512$ |
| **Transformer** | | | |
| $t$ | Transformer encoder | x6 | see Table 6 |
| $g$ | $\left\{\begin{array}{l}\text{linear layer}\\ \text{ReLU}\end{array}\right.$ | $\Big\}$x4 | each $512 \times 512$ |
| $f$ | $\left\{\begin{array}{l}\text{(Dropout)}\\ \text{linear layer}\\ \text{ReLU}\end{array}\right.$ | $\Big\}$x3 | p = 0.5 (only for second layer) $512 \times 256, 256 \times 256$ and $256 \times 13$ |

panel and logits for corresponding probabilities of the rules embedded in a PGM. The model is illustrated in Figure 2.

First, representations for context panels, denoted by $\mathbf{x}_i$, and representations for choice panels, denoted by $\mathbf{c}_k$, are generated using a shared *Convolutional Neural Network* (CNN). $\mathbf{x}_i$ denotes the $i$-th context panel feature, $\mathbf{c}_k$ denotes the $k$-th choice panel feature. This network has the same hyperparameters as the CNN in the Wild Relation Network (WReN). A one-hot positional encoding indicating one out of nine possible positions within the panel grid is concatenated to each extracted panel feature and the resulting vector is projected to obtain a final representation for each panel (see *panel encoder* and *concatenation* blocks in Figure 2).

Sequences $\boldsymbol{\chi}_k$ of length $N$ are composed of the context and choice panel representations. Optionally, a learnable deliminiter $\mathbf{d}$ which has the same number of dimensions as the panel representations can be included between contexts and choice. Thus, sequences of $N = 9$ or $N = 10$ elements are obtained:

$$\boldsymbol{\chi}_k = (\mathbf{x}_1, \mathbf{x}_2, \ldots, \mathbf{x}_8, \mathbf{d}, \mathbf{c}_k) \ . \tag{1}$$

We generate a sequence $\boldsymbol{\chi}_k$ for all eight choices. A multi-step attention network inspired by the encoder of the Transformer model $t$ (Vaswani et al., 2017), processes this sequence using the self-attention mechanism for abstract reasoning. All activations are accumulated before the network's output is generated by the MLP $f$.

$$\mathbf{o}_k = f \left( \sum_{i=1}^{N} g \left( t \left( \boldsymbol{\chi}_k \right) \right)_i \right) , \tag{2}$$

where $g$ denotes the output of the 4-layer MLP (see $f$ and $g$ decoder blocks in Figure 2). The vector $\hat{\mathbf{p}} \in \mathbb{R}^8$ denotes the logits of the corresponding probability distribution over the choices and the matrix $\hat{\mathbf{A}} \in \mathbb{R}^{8 \times 12}$ indicates the logits of the corresponding presence probabilities of rules imposed by the panels where $\hat{\mathbf{a}}_k \in \mathbb{R}^{12}$ is defined as a transposed row vector of this matrix. The construction of these meta-targets is explained subsequently.

The network returns a matrix $\mathbf{o} \in \mathbb{R}^{8 \times 13}$, which contains both, logits for the prediction of a choice panel and logits for the prediction of underlying patterns across the PGM

$$\begin{pmatrix} \hat{p}_1 & \hat{\mathbf{a}}_1^T \\ \vdots & \vdots \\ \hat{p}_8 & \hat{\mathbf{a}}_8^T \end{pmatrix} = \begin{pmatrix} \mathbf{o}_1 \\ \vdots \\ \mathbf{o}_8 \end{pmatrix} . \tag{3}$$

Table 2: PGM accuracy by previous methods and our model (ARNe). Accuracy of WReN as reported in (Santoro et al., 2017) and our implementation. MAC model included for comparison.

| Model | | Accuracy [%] |
|---|---|---|
| MAC (our implementation) | | 12.6 |
| VAE-WReN (Steenbrugge et al., 2018) | $\beta = 4$ | 64.2 |
| WReN (Santoro et al., 2017) | $\beta = 10$ | 76.9 |
| WReN (our implementation) | $\beta = 10$ | 79.0 |
| WReN-MAC | | 79.6 |
| **ARNe (our implementation)** | $\beta = 10$ | **88.2** |

**Meta-targets**  In the PGM dataset, the layout and appearance of the nine panels follow an implicit rules. Each rule is represented by a triplet containing an object type, object attribute and relation type. An example rule could be (shape, size, progression) to describe a pattern of triangles with increasing size. Up to four different relational rules (i.e. four triplets) may occur simultaneously in a single PGM. Meta-targets are created by aggregating all binary encoded rules with an OR operation resulting in a 12 dimensional meta-target vector.

In order to identify correct panels, a successful model should be able to infer the underlying construction principle reliably. To enhance rule prediction in the model, we incorporate the auxiliary information encoded in the meta-targets in the loss function. Since each choice panel results in a prediction $\hat{\mathbf{a}}_k$ of a meta-target, the predictions need to be aggregated by a sum over choice panels, implying an OR relation between the elements (Santoro et al., 2018).

**Loss**  The loss $\mathcal{L}$ is defined by a weighted combination of finding the right choice $\mathbf{p}$ and detection of the correct logical pattern $\mathbf{a}$ (Santoro et al., 2018):

$$\mathcal{L} = \text{CE}(\mathbf{p}, \hat{\mathbf{p}}) + \beta \cdot \text{BCE}(\mathbf{a}, \sum_{k=1}^{8} \hat{\mathbf{a}}_k) \, ,$$

with CE and BCE being cross entropy and binary cross entropy. The parameter $\beta$ controls the influence of the meta-targets, i.e. for $\beta = 0$ the meta-targets are ignored. By enforcing correctness of the meta-targets we enable richer gradients that identify better parameters.

**Implementation**  The learning is carried out using an adam Kingma & Ba (2014) optimizer with a batch size of $64$ and an initial learning rate of $0.5 \cdot 10^{-4}$ using a learning rate scheduler with exponential decay. We apply early stopping (Goodfellow et al., 2016) with a patience of three epochs. Parameters of the individual components of our method are presented Table 1. For further details we refer to the appendix or the implementation of our approach which is available here: `http://hidden for blind review. Will be disclosed upon acceptance`

## 4  EXPERIMENTS

We evaluate ARNe on the PGM dataset (DeepMind, 2017). Each PGM input sample consists of $160 \times 160$ pixel images. In total, there are 16 panels in every sample, eight of which define the context and the remaining eight define possible choices (see Figure 1). Similar to RPMs, a correct panel has to be chosen that matches the implicit relational rules encoded in the context panels. PGM includes meta-targets in the form of 12-bit feature vectors that denote relation, object and attribute types. The rules that underlie each sample are composed of 1 to 4 relational rules, chosen from the set (*Progression*, *AND*, *OR*, *XOR*, *Consistent Union*). Figure 1b,c shows an example of the *Progression* rule.

We trained ARNe on the $1.2 \times 10^6$ training samples from the PGM dataset where early stopping terminated training after 45 epochs. After training, ARNe detected answer panels with an accuracy of $88.18\%$ and auxiliary data with $98.72\%$. F1-score reached $0.9801$. For comparison, we included in Table 2 results for WReN (Santoro et al., 2017) and VAE-WReN (Hudson & Manning, 2018) as baseline. These models were significantly outperformed by ARNe. It is noteworthy that our re-

Table 3: Meta-target prediction performance.

| Meta-target | Accuracy [%] | Precision [%] | Recall [%] | F1-Score [%] |
|---|---|---|---|---|
| Progression | 96.53 | 94.07 | 90.52 | 92.26 |
| AND | 97.76 | 96.92 | 94.01 | 95.44 |
| OR | 98.15 | 98.53 | 93.99 | 96.21 |
| XOR | 97.59 | 99.07 | 91.23 | 94.99 |
| Consistent Union | 96.87 | 97.63 | 91.79 | 94.62 |
| Shape | 99.82 | 99.91 | 99.81 | 99.86 |
| Line | 99.98 | 99.98 | 99.98 | 99.98 |
| Size | 99.93 | 99.89 | 99.77 | 99.83 |
| Type | 99.95 | 99.91 | 99.98 | 99.94 |
| Position | 99.98 | 99.91 | 99.99 | 99.95 |
| Number | 99.97 | 99.92 | 99.84 | 99.88 |
| Color | 98.13 | 99.01 | 96.80 | 97.89 |

implementation of WReN also outperformed (Santoro et al., 2017) by 2.1 ppt despite careful code validation (see Table 2 and Appendix B).

In addition, we included a comparison with the Attention and Composition (MAC) Network (Hudson & Manning, 2018). To the best of our knowledge this is the first time MAC is tested on the PGM dataset. To our surprise MAC did not perform significantly above chance level on this task. We also tested a version of MAC that uses a WReN at the input stage, similar to ARNe. We call this augmented variant WReN-MAC. Our first results with this model suggest that also WReN-MAC does not reach the performance of ARNe. We ran tests on a reduced PGM dataset that used only 20% of the full training set. WReN-MAC achieved 46.9% test accuracy, about 10% below that of ARNe on the same dataset size (see Figure 3). We are running simulations with WReN-MAC on the full dataset but by the time of the submission of this paper these experiments were not concluded. Additional details on the implementation of WReN-MAC are provided in Appendinx D.

To gain additional insights into the learning behavior of ARNe we clustered the test dataset by the number of relational dependencies. ARNe showed good performance on all types, performing best on samples with four (accuracy: $90.54\%$) and worst for samples with three (accuracy: $82.74\%$) relational rules. Next, we evaluated the ability of ARNe to predict the meta-targets. Table 3 shows individual performances for each of the 12 meta-targets. ARNe achieved high accuracy in all categories. Detection rates revealed an unconditional accuracy rate of above $90\%$ consistently across all meta-target types.

Furthermore, we evaluate the generalization abilities of ARNe by the provided samples in the PGM dataset. Consistently with the scores on the neutral split, we find our method to outperform the wild relation network in Tab. 4.

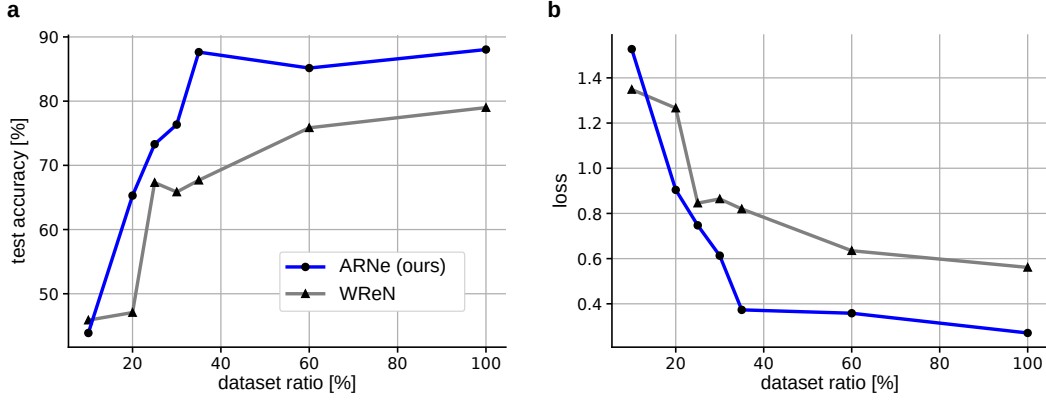

Figure 3: **Sample efficiency:** Test accuracy (**a**) and loss (**b**) after training on different numbers of samples (percent of original dataset size). Our implementation of WReN was used here (see Appendix B).

Table 4: Generalisation results. No training results for WReN were available (Santoro et al., 2018). Values in the training, validation and test columns characterise the model's accuracy.

| Model | $\beta$ | Training [%] | Validation [%] | Test [%] |
|---|---|---|---|---|
| WReN (Santoro et al., 2018) | 10 | - | 93.60 | 15.50 |
| **ARNe** | **10** | **99.43** | **98.93** | **17.76** |

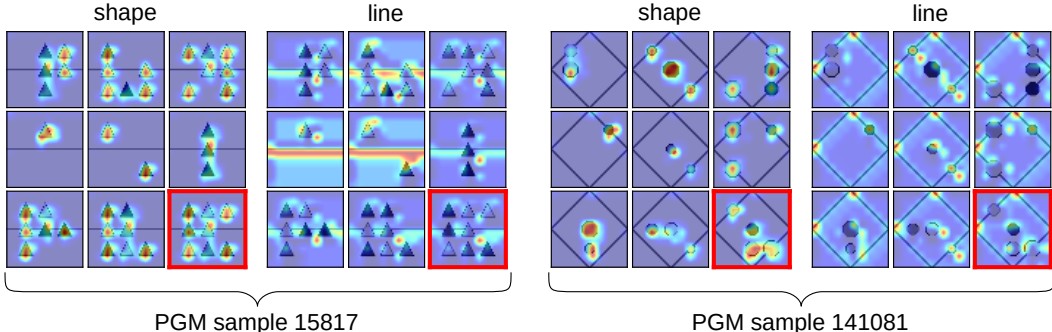

Figure 4: Sum of selected activations of the last layer of the panel encoder CNN, that encoded predominantly for shapes (maps 14, 24, 26, 29, 31) and for lines (maps 1, 2, 3, 4, 8, 12). Matrices show complete sets of context panels and the correct choice panel highlighted in red.

## 4.1 SAMPLE EFFICIENCY

In order to evaluate the sample efficiency of ARNe, we trained the model subsequently with growing fractions of the training dataset. The validation and test sets were kept unchanged during the entire experiment. Prior to splitting of the training set, the dataset was shuffled and the random seed was kept fixed throughout all experiments. The results are presented in Figure 3. ARNe achieved higher test accuracy across all dataset ratios $\geq 20\%$. Using a split size of $35\%$, the accuracy reads $87.64\%$, already close to the value when trained on the full training set and $8.65$ ppt better than the WReN model at $100\%$ of the training dataset. The progression of the loss shows a similar behavior (Figure 3b).

## 4.2 SELF-ATTENTION FOR VISUAL REASONING

In order to gain insights into how the self-attention mechanism of ARNe works internally we visualize the activations of the panel encoder for different encodings of the meta-targets in Figure 4. It is important to note that these features captures all relevant information for reasoning about relational rules between objects. Therefore, recognizing objects of specific types is an important sub-task for deciding which objects are relevant for a particular relational rule. Figure 4 shows two examples for the *object type* meta-targets *shape* and *line*. For each object type, output channels of the panel encoder CNN were selected that strongly responded. The eight choices and the corresponding correct context panel in the bottom right, are shown.

We found that, after training, many panel encoder CNN output channels significantly encoded activation maps for a single meta-target type. The activations in Figure 4 most saliently select regions within the input panels that are part of line or shapes objects.

## 4.3 ABLATION

We experimented with various configurations of ARNe. This involves dropout to reduce over-fitting, a different learning rate, the inclusion of the delimiter token when assembling sequences and the weighting of meta-targets $\beta$. Table 5 depicts the results of the subsequent and more detailed ablation studies of ARNe. Further values are as listed in the Appendix A. For small dataset dropout gives a substantial performance boost, while this is not the case when the full dataset is used. This is likely due to the additional samples preventing the network from over-fitting. Similarly, the addition

Table 5: Comparison of different model configurations. The highlighted lines refer to the best performing model of the given dataset ratio. Values in the training, validation and test columns characterize the model's accuracy. (lr = learning rate).

| Dropout [%] | lr$\times 10^{-4}$ | Delimiter | $\beta$ | Ratio [%] | Train [%] | Val. [%] | Test [%] |
|---|---|---|---|---|---|---|---|
| 10 | 0.5 | ✗ | 10 | 35 | 81.83 | 78.67 | 78.32 |
| 10 | 0.5 | ✓ | 10 | 35 | 82.21 | 79.39 | 78.69 |
| 17 | 0.5 | ✗ | 10 | 35 | 93.74 | 85.10 | 84.35 |
| **17** | **0.5** | **✓** | **10** | **35** | **93.83** | **87.65** | **87.64** |
| 10 | 0.5 | ✗ | 0 | 100 | 12.50 | 12.65 | 12.55 |
| 17 | 0.5 | ✗ | 10 | 100 | 86.27 | 87.72 | 87.11 |
| 10 | 1 | ✗ | 10 | 100 | 88.86 | 88.66 | 87.95 |
| 17 | 0.5 | ✓ | 10 | 100 | 88.23 | 88.42 | 88.04 |
| **10** | **0.5** | **✗** | **10** | **100** | **89.06** | **88.77** | **88.18** |
| Feed-forward net instead of Encoder | | | | | 32.02 | 35.87 | 35.06 |
| Feed-forward net instead of Self-attention | | | | | 38.79 | 45.12 | 44.56 |

of a deliminiter tends to improve performance for small datasets, thus it might act as a regularizer. Furthermore, we find that the method is robust to a changed initial learning rate.

An intriguing insight is that ARNe does not converge when no meta-targets are provided ($\beta = 0$). This indicates the relevance of the auxiliary signals which connect the panel choice with the underlying implicit relational rules during training (at test no meta-targets were presented). Beyond the findings of Table 5, we investigated improvements of components of WReN as well as the addition of FiLM Perez et al. (2018) layers but did not notice a performance improvement. However, by reducing the input image size from $160 \times 160$ to $80 \times 80$ pixels, accuracy decreased by about 15 ppt. This suggests that relevant small structures exist and are used by the model to solve the PGM.

### 4.4 RAVEN DATASET

For comparison, we also evaluated our model on the recently published RAVEN dataset Zhang et al. (2019). We found it achieve a performance of 92.23% for 50,000 samples for each figure configuration. However, using Raven-10000, i.e. using only 10,000 samples, the performance dropped to 19.67%.

## 5 CONCLUSION

In this work we introduced ARNe, a new deep learning model that combines features from the Wild Relation Network (WReN) and the Transformer network to discover patterns in progressive matrix panels using aspects of fluid intelligence. More precisely, ARNe builds upon the WReN and extends it with the attention mechanism of the Transformer, which originates from language modelling, making it to our knowledge the first deep learning approach that uses self-attention specifically for abstract visual reasoning. The learning is driven by an auxiliary loss that gives hints about the underlying patterns of the progressive matrix panels. In an extensive experimental comparison we find that ARNe outperforms state-of-the-art abstract reasoning methods on the PGM dataset by a large margin. Moreover, the analysis shows that our model is substantially more sample-efficient than competing approaches.

Our experiments also including the first application of MAC to the PGM dataset. But the MAC did not seem to be well suited for this task, and also showed significantly worse performance compared to baseline, when it was augmented with a WReN-based panel encoder. Additional experiments are needed to gain a deeper understanding on why MAC appears to fail here. First insights into the inner workings of ARNe are given in Figure 4. We show that the panel encoder CNN generates meaningful features that aid the learning goal. This suggests that the WReN and Transformer network parts of ARNe learn to efficiently cooperate for abstract reasoning.

Altogether, these results suggests that the self-attention mechanism can be helpful in domains beyond text processing. Future work involves modifying our method such that it can be used on visual reasoning datasets which require a different structure, e.g. VisualQA datasets which involve parsing an explicit question.

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

# APPENDIX

## A    DETAILS TO ARNE IMPLEMENTATION

The model was trained using a learning rate of $0.5 \cdot 10^{-4}$. A heuristic learning rate scheduler was employed which is based on the official MAC implementation (Drew A. Hudson, 2018). It triggers below a loss of $0.6$ and utilizes a decay parameter of $0.75$. Early stopping (Goodfellow et al., 2016) was used which stopped training after three subsequent epochs of no improvements of the model's validation loss. For optimization, Adam (Kingma & Ba, 2014) was used. Parameters of ARNe's Transformer encoder are listed in Table 6. We used our own PyTorch implementation of the Transformer since at the beginning of this project the reference implementation[1] was not available.

Table 6: Model parameters of the Transformer encoder used in the conducted experiments.

| Dropout Layer | $p_{drop}$ [%] |
|---|---|
| $drop_{attention}$ | 10 |
| $drop_{position}$ | 10 |
| **Parameter** | **value** |
| $d_{model}$ | 512 |
| $d_k$ | 64 |
| $d_q$ | 64 |
| $d_v$ | 64 |
| h | 10 |
| $N_{layers}$ | 6 |
| $d_{hidden}$ | 2056 |
| **Linear Layer** | **value** |
| $linear_v$ | $[d_{model}, h \cdot d_v]$ |
| $linear_k$ | $[d_{model}, h \cdot d_k]$ |
| $linear_q$ | $[d_{model}, h \cdot d_q]$ |
| fc | $[h \cdot d_k, d_{model}]$ |
| **Convolutional Layers** | $[d_{input}, d_{output}, \text{kernel, stride, padding}]$ |
| $FC_{position}$ | $[d_{model}, d_{hidden}, 1, 1, 0]$ |
| | ReLU |
| | $[d_{hidden}, d_{model}, 1, 1, 0]$ |

## B    DETAILS TO WRENE IMPLEMENTATION

The WReN model was re-implemented in PyTorch as described in (Santoro et al., 2018) and tested on the neutral PGM dataset (DeepMind, 2017). The model was trained with $\beta = 10$ only. A deviating batch size of $64$ instead of $32$ was used. This yields accuracy rates of $78.49\%$, $80.02\%$ and $79.00\%$ for training, validation and testing respectively. This implies an improvement of $2.82$ ppt for validation and $2.1$ ppt for testing compared to the WReN baseline model (Santoro et al., 2018). Figure 5 displays the progression of accuracy and loss during training, validation and testing.

## C    DETAILS TO MAC IMPLEMENTATION

The MAC network was implemented as reported in (Hudson & Manning, 2018) using PyTorch. The implementation was verified on the CLEVR dataset. Loss and accuracies of our implementation

---

[1] https://github.com/pytorch/pytorch/releases/tag/v1.2.0

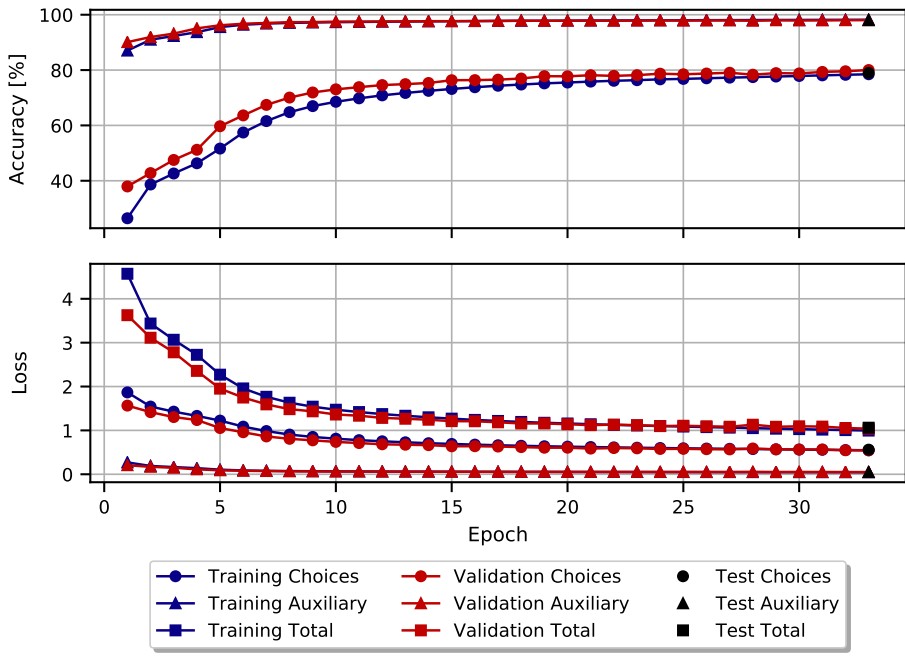

Figure 5: *Accuracy and Loss of the self implemented WReN (Santoro et al., 2018) model for $\beta = 10$ on the neural PGM dataset. The development of both metrics include the progression of the panel choices, additional auxiliary data which describes the setup of the PG-Matrices. In addition, the loss shows the overall sum.*

of MAC on the CLEVER dataset can be seen in Figure 6. Best value for validation regarding the accuracy is $97.83\%$. The training, validation demanded a computation effort of about half a week on a GeForce GTX TITAN X graphics card.

## D    DETAILS TO WREN-MAC IMPLEMENTATION

The WReN-MAC model uses the same encoding mechanism as the WReN baseline model and includes 12 MAC-cells. In order to interface a MAC-cell, additional convolutional layers were appended to the adapted WReN convolutional network. The model is analogously sequentially aligned like WReN or WReN-Transformer whereas during each pass one knowledge base which encodes one distinct choice panel is used. The question vector $\vec{q}$ is computed in analogy to WReN where $g_\theta$ and the subsequently applied sums were used. The computation of the model's loss also respected auxiliary structure set data of the corresponding PGMs.

We were not able to train WReN-MAC successfully on the PGM dataset. Early stopping finished learning after 33 epochs. The final accuracies and losses are $49.26\%$ and $3.04$, $46.97\%$ and $3.20$, $46.89\%$ and $3.23$ for training, validation and testing respectively. Both accuracy and loss showed marginal improvements throughout all epochs. Due the recurrent nature of MAC-cells, we used gradient clipping (Goodfellow et al., 2016). The learning rate scheduler was set but didn't diminished the the learning rate which was initially set to $1 \cdot 10^{-4}$. Additional hyperparameter tuning did not show significant performance increase.

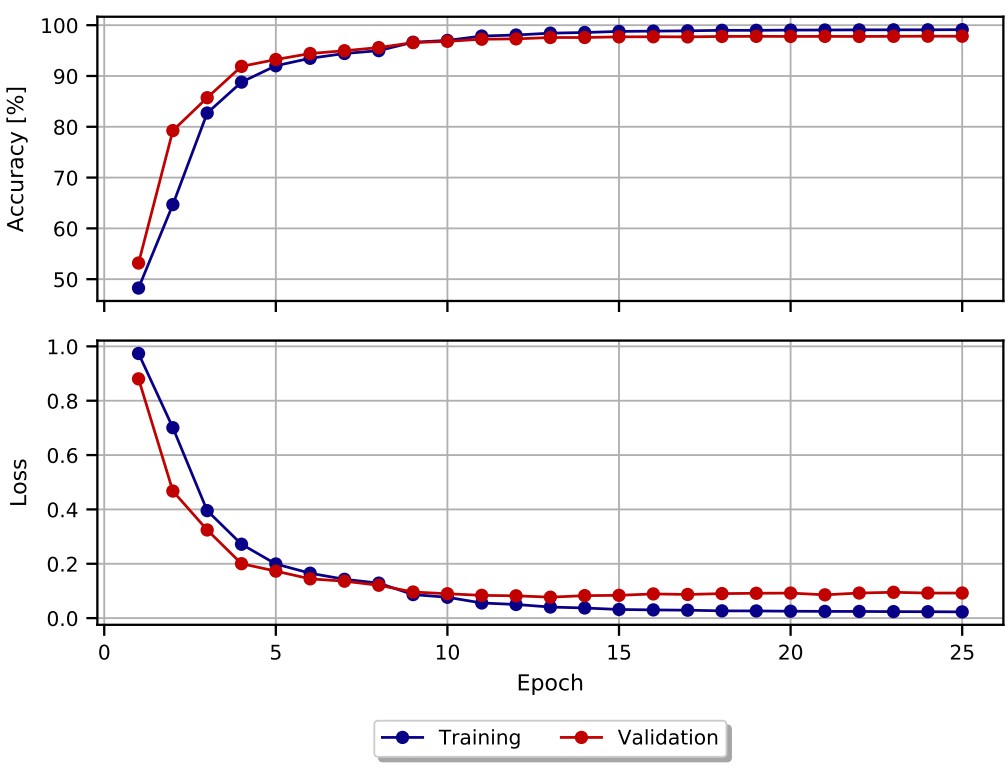

Figure 6: Learning curves during training and validation of our implementation of MAC on the CLEVR dataset. **Accuracy**: Final learning rate values after 25 epochs are 97.83% and 99.11% for the validation and training set respectively. **Loss**: Calculated losses of the implemented neural network with respect to epochs. Final loss values after 25 epochs are $92.30 \cdot 10^{-3}$ and $22.90 \cdot 10^{-3}$ for the validation and training set respectively.

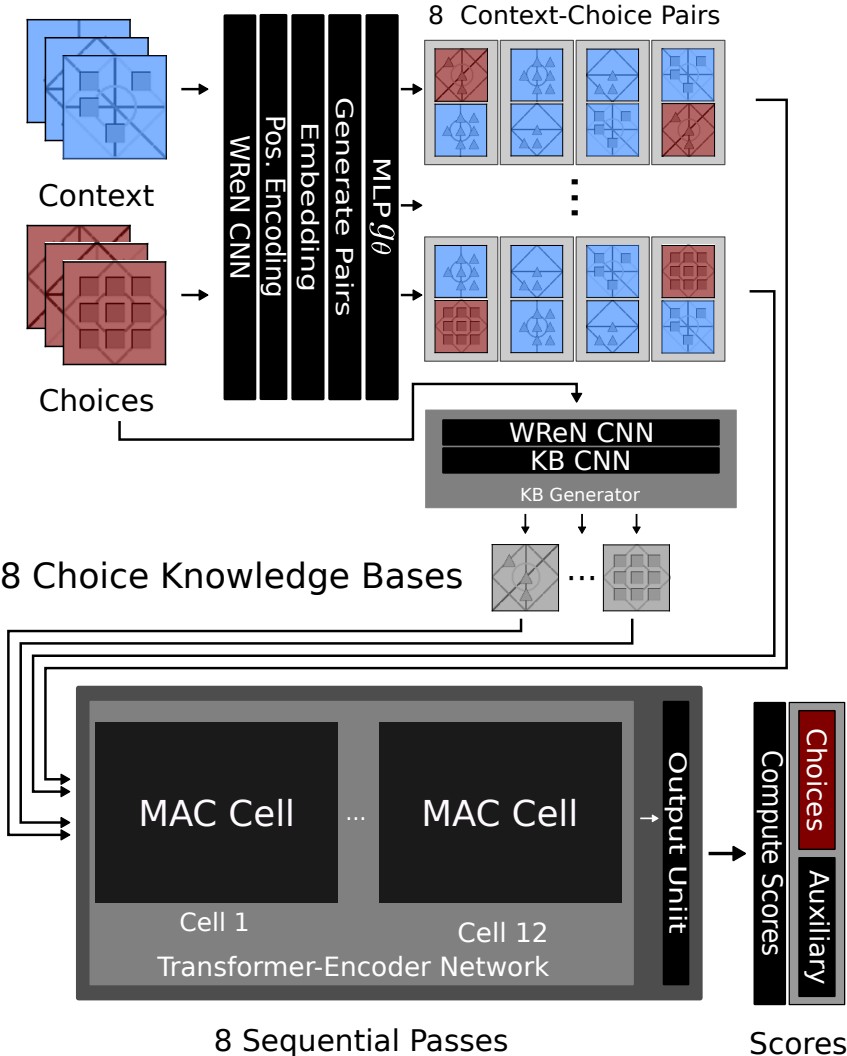

Figure 7: WReN-MAC model. First, eight sequences of embeddings and $g_\theta$ activations are generated analogously to WReN. To interface a MAC-cell properly, the knowledge base is required. A knowledge base is generated by the WReN CNN plus additional layers to match MAC's required dimensions. Every sequence passes 12 recurrent MAC-cells and the output unit sequentially. The computation of scores equals the method incorporated in the WReN model.

