# OpenReview forum: "Attention on Abstract Visual Reasoning"
_ICLR.cc/2020/Conference — Reject_

### Official Review · AnonReviewer2 · 2019-10-22
**Official Blind Review #2**

**Rating:** 3

**Review:**

This work introduced an attention-based model to solve the RPM cognitive tasks. The model is based on the transformer network, which performs relational reasoning through its self-attention mechanisms.

Technical novelty:
The method seems to be a straightforward application of the transformer network to the PGM task. The technical novelty of the proposed approach is unclear. I’d love to hear what the authors have to say about the technical contributions of the proposed ARNe model in comparison to prior work.

Supervision with meta-targets:
It also seems that the meta-targets are crucial for attaining a good performance with the ARNe model. According to Table 4, the model without meta-target training (beta=0) only achieved 12% accuracy in train/val/test sets. However, prior work [Santoro* et al. 2018] has demonstrated that even without training on meta-targets, WReN still achieves a performance of over 60% accuracy (Table 1). This result suggests that the proposed ARNe model does not work well when training with weaker supervision without meta-targets. The results could be a lot stronger if the authors show ARNe outperforms the prior work when beta is set to 0.

Ablation studies:
This model is only tested in the neutral PGM dataset. The evaluation would be strengthed with the generalization results of this model in different generalization regimes (see Table 1, Santoro* et al. 2018) and comparing its performance with prior works.

**Experience Assessment:**

I have read many papers in this area.

**Review Assessment: Checking Correctness Of Derivations And Theory:**

I carefully checked the derivations and theory.

**Review Assessment: Checking Correctness Of Experiments:**

I carefully checked the experiments.

**Review Assessment: Thoroughness In Paper Reading:**

I read the paper at least twice and used my best judgement in assessing the paper.

---

### Official Review · AnonReviewer3 · 2019-10-22
**Official Blind Review #3**

**Rating:** 3

**Review:**

This paper describes a somewhat novel approach to abstract visual reasoning using transformers in the so-called "Attention Relation Network" (ARNe), which the authors show to improve on the "Wild Relation Network" (WReN). The Transformer is motivated by the role that attention may play in Human information processing - which sounds plausible, but the paper does not expand on this theme.

The paper is well written and makes an interesting contribution, but I feel the results are not quite yet ready for publication. The authors are writing that they are still working on baseline results on the full dataset, which would provide interesting comparisons, and some details on the implementation (number of parameters, etc) are missing - or maybe I missed them.

The learning curve in Figure 3 (sample efficiency, test accuracy) suggests that the ARNe training is not fully stable - why would the model deteriorate when going from ~40% of the training data to ~60%? Is the model potentially overfitting, and how does the size of the proposed model compare to the size of the baseline model(s)? It seems that the field is also moving towards the RAVEN dataset, which presents a more complex structure; it would be more convincing to present results on both datasets, to show that attention can indeed also improve results on more complex setups.

The text in the "Acknowledgments" section should be removed for the camera ready version!


**Experience Assessment:**

I do not know much about this area.

**Review Assessment: Checking Correctness Of Derivations And Theory:**

I did not assess the derivations or theory.

**Review Assessment: Checking Correctness Of Experiments:**

I did not assess the experiments.

**Review Assessment: Thoroughness In Paper Reading:**

I made a quick assessment of this paper.

---

### Official Review · AnonReviewer1 · 2019-10-24
**Official Blind Review #1**

**Rating:** 1

**Review:**

This work proposes a new architecture for abstract visual reasoning, based on Transformer-style soft attention and relation networks. The authors test their network on the PGM dataset, and demonstrate a non-trivial improvement over previously reported baselines.

In general, abstract reasoning is an important field of current study in neural network-based machine learning, as it is an area that has notoriously eluded these types of models historically. The paper is reasonably well put together, and I have no reason to question the various technical aspects of the work.

Unfortunately, I think there are significant shortcomings. Firstly, the PGM dataset was designed to stress out-of-distribution generalization, and performance on the Neutral split was not proposed as a particularly interesting challenge on its own. This is because, as the name implies, abstract reasoning requires the ability to identify abstract conceptual features of the data and compose them in novel ways at test time, which is *not* a feature of the neutral split.  The authors are encouraged to run their model on these other generalization splits.

Second, there seems to be little value to the field overall for research involving minor architectural improvements for single datasets. If the authors believe in this method, they are encouraged to demonstrate its effectiveness on a wide variety of data types. On this note, I should add that the authors are incorrect to state that this is the first work to use self-attention for abstract reasoning (please see Zambaldi, 2018 for one example of many papers that have incorporated self-attention into convolutional architectures).

So to sum up, while this work broaches an interesting subject and is technically fine, it does not surpass the threshold for acceptance because it fails to demonstrate the usefulness of the method on the task at hand, as well as broad utility of the proposed method.


**Experience Assessment:**

I have published in this field for several years.

**Review Assessment: Checking Correctness Of Derivations And Theory:**

N/A

**Review Assessment: Checking Correctness Of Experiments:**

I carefully checked the experiments.

**Review Assessment: Thoroughness In Paper Reading:**

I read the paper thoroughly.

---

### Public Comment · ~Hyunjae_Kim1 · 2019-10-05
**Several concerns about your paper**

The proposed model, which is a transformer-based RPM solver, achieved significantly high accuracy in the neutral setting, where the data distribution of training and test sets are the same.
I think the idea of adopting the attention mechanism in solving abstract reasoning tasks is good.


However, the model was not tested on generalization settings such as H.O. Triple Pairs, Interpolation and Extrapolation, where unseen objects and attributes appear during evaluation (Barret et al. 2018).
Since the PGM dataset was proposed to evaluate the generalization abilities of models, I believe evaluation should have been conducted not only on the neutral setting, but also the generalization settings.

Also, there is an another benchmark dataset, called RAVEN (Zhang et al. 2019).
It would be better if the proposed model was evaluated on both PGM and RAVEN.

Lastly, I am concerned that the ablation study in the paper is insufficient.
It is questionable whether the high accuracy in the neutral setting is due to the effectiveness of the self-attention mechanism or just the large model size.

---

> ### Author Response · Authors · 2019-10-23
> **Further experiments on PGM and RAVEN**
>
> Thank you for your comment. In the following, we reply to each criticized point.
>
> Generalization:
> We agree that this would enable further insights. We have tested it on the provided extrapolation dataset. ARNe achieved a performance of 17.76% which is slightly better than WReN. Unfortunately, we do not have the resources to conduct experiments of other PGM configurations: They would require large store capacities as well as extensive computations.
>
> RAVEN benchmark:
> Thank you for pointing out this new benchmark. We evaluated our model on RAVEN and found it achieve a performance of 92.23% for 50k samples for each figure configuration. However, using Raven-10000 the performance is 19.67%.
>
> Ablation: We replaced the encoder with a MLP of the same depth. In a second experiment we replaced the multi head attention mechanism with a linear transformation. The performances are 35.06% and 44.56% respectively.
>
> To further strengthen the experimental validation of our model, we implemented a combination of WReN and MAC called WReN-MAC and found it achieve 79.6 % on PGM.
>
> The paper will be updated to reflect these additional findings.

---

### Author Response · Authors · 2019-11-15
**Reply to Reviewers**

We thank the reviewers for their comments and suggestions. Based on the reviews, we made the following changes to the paper:
- We added an evaluation of the model's performance on the extrapolation split.
- We conducted experiments on the new RAVEN dataset and report the results

Response to the key criticism:
- Of course, the dependency of the ARNe model on labeled data is a limitation. However, this requirement only affects training. At test time, the model does not need auxiliary labels.
- The neutral split of the PGM dataset was used by [1] to compare with other models. Therefore the argument that the neutral split was not intended to be an interesting challenge seems misplaced. Nonetheless, the performance on other splits is interesting. Hence, we added results on the extrapolation split (more splits were not possible in this rebuttal period due to time constraints).
- We agree that additional datasets could strengthen the paper. A small experiment on RAVEN was added.

[1] Santoro et al., 2018: Measuring abstract reasoning in neural networks

---

### Decision · Program_Chairs · 2019-12-19

**Decision:**

Reject

**Comment:**

This work proposes a new architecture for abstract visual reasoning called "Attention Relation Network" (ARNe), based on Transformer-style soft attention and relation networks, which the authors show to improve on the "Wild Relation Network" (WReN). The authors test their network on the PGM dataset, and demonstrate a non-trivial improvement over previously reported baselines.

The paper is well written and makes an interesting contribution, but the reviewers expressed some criticisms, including technical novelty, unfinished experiments (and lack of experimental details), and somewhat weak experimental results, which suggest that the proposed ARNe model does not work well when training with weaker supervision without meta-targets. Even though the authors addressed some concerns in their revised version (namely, they added new experiments in the extrapolation split of PGM and experiments on the new RAVEN dataset), I feel the paper is not yet ready for publication at ICLR.